# Learning Heuristics for Quantified Boolean Formulas through Reinforcement Learning

**Gil Lederman**
Electrical Engineering and Computer Sciences
University of California at Berkeley
gilled@eecs.berkeley.edu

**Markus N. Rabe**
Google Research
mrabe@google.com

**Edward A. Lee**
Electrical Engineering and Computer Sciences
University of California at Berkeley
eal@eecs.berkeley.edu

**Sanjit A. Seshia**
Electrical Engineering and Computer Sciences
University of California at Berkeley
sseshia@eecs.berkeley.edu

## Abstract

We demonstrate how to learn efficient heuristics for automated reasoning algorithms for quantified Boolean formulas through deep reinforcement learning. We focus on a backtracking search algorithm, which can already solve formulas of impressive size - up to hundreds of thousands of variables. The main challenge is to find a representation of these formulas that lends itself to making predictions in a scalable way. For a family of challenging problems in 2QBF we learn a heuristic that solves significantly more formulas compared to the existing handwritten heuristics.

## 1 Introduction

One of the most intriguing questions for artificial intelligence is: can (deep) learning be effectively used for symbolic reasoning? The benefits of combining deductive reasoning with inductive learning for automated reasoning and in formal methods for system design have been noted (e.g., see Seshia (2015)). There is a whole spectrum of approaches to combine them: One extreme is to use learning for predicting which of a small pool of algorithms (or heuristics) performs best, and run only that one to solve the given problem (e.g. SATzilla (Xu et al., 2008)). This approach is clearly limited by the availability of handwritten algorithms and heuristics (i.e. it can only solve problems for which we have written at least one algorithm that can solve it). The other extreme is to analyze formulas solely with deep learning approaches (Allamanis et al., 2016; Evans et al., 2018; Selsam et al., 2018; Amizadeh et al., 2019). However, this approach shows poor scalability compared to the state-of-the-art in the respective domains depsite the recent breakthroughs in deep learning. Instead of relying entirely on deep learning or on the availability of good handwritten algorithms, we explore the middle ground. We ask the question how to tightly combine deep learning with formal reasoning algorithms with the goal to improve the state-of-the-art, i.e. to solve formulas that could not be solved previously.

Existing formal reasoning tools work in a mechanical way: they only apply a small number of carefully crafted operations and use heuristics to resolve the degrees of freedom in how to apply them. We address the problem of automatically learning better heuristics for a given set of formulas. We focus on the branching heuristic in modern backtracking search algorithms, as they are known to have a high impact on the performance of the algorithm. We cast the problem to learn better branching heuristics for backtracking search algorithms as a reinforcement learning problem: Initially, the reinforcement learning environment randomly picks a formula from a given set of formulas, and then runs the backtracking search algorithm on that formula. The actions that are controlled by the learning agent are the branching decisions, i.e. pick a variable and assign it a value - everything else is handled by the solver.

**Challenges** This reinforcement learning problem comes with several unique challenges:

*Representation:* While learning algorithms for images and board-games usually rely on the grid-like structure of the input and employ neural networks that match that structure (e.g. convolutional neural networks). For formulas, however, there is no standard representation for learning algorithms.

It may seem reasonable to treat Boolean formulas as text and learn embeddings for formulas through techniques such as word2vec (Mikolov et al., 2013), LSTMs, or tree RNNs. However, formulas in formal reasoning tools typically consist of thousands of variables, which is much larger than the text-fragments typically analyzed with neural networks. Further, unlike words in natural language, individual variables in Boolean formulas are completely devoid of meaning. The meaning of variable $x$ in one formula is basically independent from variable $x$ in a second formula. Hence, learning embeddings for variables and sharing them between formulas would be futile.

*Unbounded action space:* An action consists of a choice of variable and value. While values will be Boolean, the number of variables depends on the size of the input formula. Therefore, we have an unbounded number of actions, which are further different for every formula.

*Length of episodes:* As we are dealing with a highly complex search problem, solver runs (= learning episodes) can be very long—in fact, for many of the formulas we have never observed a terminating run—and we observed a huge variance in the length of runs.

*Performance:* Our aim is to solve more formulas in less time. The use of neural networks incurs a huge runtime overhead for each decision (the solver takes $\geq$10x fewer decisions per second). So the decisions taken by the neural networks need to be dramatically better than the handcoded heuristic decisions to outweigh their runtime cost.

*Correctness:* Reinforcement learning algorithms have shown to often find and exploit subtle implementation errors in the environment, instead of solving the intended problem. While testing and manual inspection of the results is a feasible approach for board games and Atari games, it is neither possible nor sufficient in large-scale formal reasoning - a solver run is simply too large to inspect manually and even tiny mistake can invalidate the result. In order to ensure correctness, we need an environment with the ability to produce formal proofs, and check the proofs by an independent tool.

**Quantified Boolean Formulas** In this paper we focus on 2QBF, that is quantified Boolean formulas of the form $\forall X.\exists Y.\varphi$, where $X$ and $Y$ are sets of Boolean variables and $\varphi$ is in conjunctive normal form. 2QBFs are complex enough to serve as an interesting proxy for complex mathematical reasoning tasks. Challenging applications such as program synthesis and the synthesis of controllers and ranking functions have been encoded into 2QBFs Solar-Lezama et al. (2006); Faymonville et al. (2017); Cook et al. (2013). However, the problem definition and also the syntactical structure of 2QBF is simple compared to more general settings, such as first-order or even higher-order logics. This makes algorithms for 2QBF a good target for the study of neural architectures.

While our approach in principle works with most algorithms for QBF, we decided to demonstrate its use in Incremental Determinization (Rabe & Seshia, 2016). We modified CADET, an open-source implementation of Incremental Determinization that performed competitively in recent QBF competitions (Pulina, 2016) and turned it into a reinforcement learning environment. The advantage of CADET in the context of reinforcement learning is its ability to produce proofs (which most other solvers do not), which ensures that the reinforcement learning cannot simply learn to exploit bugs in the environment.

**Graph Neural Networks** We consider each constraint and each variable of a given formula as a node in a graph. Whenever a variable occurs in a constraint, we draw an edge between their nodes. We then use a Graph Neural Network (GNN) (Scarselli et al., 2009) to predict the quality of each variable as a decision variable, and pick our next action accordingly. GNNs allow us to compute an embedding for every variable, based on the occurrences of that variable *in the given formula*, instead of learning an embedding that is shared across all formulas. Based on this embedding, we then use a policy network to predict the *quality* of each variable (or literal), and choose the next action accordingly. GNNs also allow us to scale to arbitrarily large formulas with a small and constant number of parameters.

**Contributions** This paper presents the successful integration of GNNs in a modern automated reasoning algorithm in a reinforcement learning setup.[1] Our approach balances the performance penalty incured by the use of neural networks with the impact that improved heuristic decisions have on the overall reasoning capabilities. The branching heuristic that we learn significantly improves CADET's reasoning capabilities on the test set of the benchmark, i.e. it solves more formulas within the same resource constraints. This is a huge step towards replacing VSIDS, the dominant branching heuristic in CDCL-based solvers for the last 20 years Moskewicz et al. (2001b); Eén & Sörensson (2003); Biere et al. (2009); Lonsing & Biere (2010); Rabe & Seshia (2016).

We also study the generalization properties of our approach: We show that training a heuristic on small and easy formulas helps us to solve much larger and harder formulas; generalization to formulas from different benchmarks is still limited though. Further, we provide an open-source learning environment for reasoning in quantified Boolean formulas. The environment includes the ability to verify its own runs, and thereby ensures that the reinforcement learning agent does not only learn to exploit implementation errors of the environment.

*Structure:* After a primer on Boolean logics in Section 2 we define the problem in Section 3, and describe the network architecture in Section 4. We describe our experiments in Section 5, discuss related work in Section 6 and present our conclusions in Section 7.

## 2 BOOLEAN LOGICS AND SEARCH ALGORITHMS

We start with describing *propositional* (i.e. quantifier-free) Boolean logic. Propositional Boolean logic allows us to use the constants 0 (false) and 1 (true), variables, and the standard Boolean operators like $\wedge$ (*"and"*), $\vee$ (*"or"*), and $\neg$ (*"not"*).

A *literal* of variable $v$ is either the variable itself or its negation $\neg v$. By $\bar{l}$ we denote the logical negation of literal $l$. We call a disjunction of literals a *clause* and say that a formula is in *conjunctive normal form* (CNF), if it is a conjunction of clauses. For example, $(x \vee y) \wedge (\neg x \vee y)$ is in CNF. It is well known that any Boolean formula can be transformed into CNF. It is less well known that this increases the size only linearly, if we allow the transformation to introduce additional variables (Tseitin, 1968). We thus assume that all formulas in this work are given in CNF.

**DPLL and CDCL.** The satisfiability problem of propositional Boolean logics (SAT) is to find a satisfying assignment for a given Boolean formula or to determine that there is no such assignment. SAT is the prototypical NP-complete problem and many other problems in NP can be easily reduced to it. The first backtracking search algorithms for SAT are attributed to Davis, Putnam, Logemann, and Loveland (DPLL) (Davis & Putnam, 1960; Davis et al., 1962). Backtracking search algorithms gradually extend a partial assignment until it becomes a satisfying assignment, or until a *conflict* is reached. A conflict is reached when the current partial assignment violates one of the clauses and hence cannot be completed to a satisfying assignment. In case of a conflict, the search has to backtrack and continue in a different part of the search tree.

*Conflict-driven clause learning* (CDCL) is a significant improvement over DPLL due to Marques-Silva and Sakallah (Marques-Silva & Sakallah, 1997). CDCL combines backtracking search with *clause learning*. While DPLL simply backtracks out of conflicts, CDCL "analyzes" the conflict by performing a couple of *resolution* steps. Resolution is an operation that takes two existing clauses $(l_1 \vee \cdots \vee l_n)$ and $(l_1' \vee \cdots \vee l_n')$ that contain a pair of complementary literals $l_1 = \neg l_1'$, and derives the clause $(l_2 \vee \cdots \vee l_n \vee l_2' \vee \cdots \vee l_n')$. Conflict analysis adds new clauses over time, which cuts off large parts of the search space and thereby speeds up the search process.

Since the introduction of CDCL in 1997, countless refinements of CDCL have been explored and clever data structures improved its efficiency significantly (Moskewicz et al., 2001a; Eén & Sörensson, 2003; Goldberg & Novikov, 2007). Today, the top-performing SAT solvers, such as Lingeling (Biere, 2010), Crypominisat (Soos, 2014), Glucose (Audemard & Simon, 2014), and MapleSAT (Liang et al., 2016), all rely on CDCL and they solve formulas with millions of variables for industrial applications such as bounded model checking (Biere et al., 2003).

---

[1]An earlier version of this work was published on `https://arxiv.org/abs/1807.08058`

**Quantified Boolean Formulas.** QBF extends propositional Boolean logic by *quantifiers*, which are statements of the form "for all $x$" ($\forall x$) and "there is an $x$" ($\exists x$). The formula $\forall x.\ \varphi$ is true if, and only if, $\varphi$ is true if $x$ is replaced by 0 (false) and also if $x$ is replaced by 1 (true). The semantics of $\exists$ arises from $\exists x.\ \varphi = \neg \forall x.\ \neg \varphi$. We say that a QBF is in prenex normal form if all quantifiers are in the beginning of the formula. W.l.o.g., we will only consider QBF that are in prenex normal form and whose propositional part is in CNF. Further, we assume that for every variable in the formula there is exactly one quantifier in the prefix. An example QBF in prenex CNF is $\forall x.\ \exists y.\ (x \vee y) \wedge (\neg x \vee y)$.

We focus on 2QBF, a subset of QBF that admits only one quantifier alternation. W.l.o.g. we can assume that the quantifier prefix of formulas in 2QBF consists of a sequence of universal quantifiers $\forall x_1 \ldots \forall x_n$, followed by a sequence of existential quantifiers $\exists y_1 \ldots \exists y_m$. While 2QBF is less powerful than QBF, we can encode many interesting applications from verification and synthesis, e.g. program synthesis (Solar-Lezama et al., 2006; Alur et al., 2013). Solvers for (2)QBF typically address the decision problem to determine the truth of a given formula (TQBF). After the success of CDCL for SAT, CDCL-like algorithms have been explored for QBF as well (Giunchiglia et al., 2001; Lonsing & Biere, 2010; Rabe & Seshia, 2016; Rabe et al., 2018). We focus on CADET, a solver that implements *Incremental Determinization* a generalized CDCL backtracking search algorithm (Rabe & Seshia, 2016; Rabe et al., 2018). Instead of considering only Booleans as values, the Incremental Determinization algorithm assigns and propagates on the level of Skolem functions. For the purpose of this work, however, we do not have to dive into the details of Incremental Determinization and can consider it simply as some backtracking algorithm.

**Correctness.** Writing performant code is an error-prone task, and correctness is critical for many applications of formal reasoning. Some automated reasoning tools hence have the ability to produce proofs, which can be checked independently. CADET is one of the few QBF solvers that can produce proofs without runtime overhead. We believe that the ability to verify results of solvers is particularly crucial for learning applications, as it allows us to ensure that the reinforcement learning algorithm does not simply exploit implementation error (bugs) in the environment.

## 3 PROBLEM DEFINITION

In this section, we first revisit reinforcement learning and explain how it maps to the setting of logic solvers. In reinforcement learning, we consider an agent that interacts with an environment $\mathcal{E}$ which is modeled as a Markov Decision Process (MDP) over discrete time steps and accumulates reward. Formally, a MDP is a 4-tuple of states $\mathcal{S}$, action space $\mathcal{A}$, transition probabilities $\mathcal{P}$ and reward function $\mathcal{R}$. A *policy* is a mapping from states to probability distributions over the actions $\pi : \mathcal{S} \rightarrow dist(\mathcal{A})$. The goal of the agent is to maximize the expected (possibly discounted) reward accumulated over the episode; formally $J(\pi) = \mathbb{E}\left[\sum_{t=0}^{\infty} \gamma^t r_t | \pi\right]$.

In our setting, the environment $\mathcal{E}$ is the solver CADET (Rabe & Seshia, 2016). The environment is deterministic except for the initial state, where a formula is chosen randomly from a distribution. At each time step, the agent gets an observation, which consists of the formula and the solver state. Only those variables that do not have a value yet are valid actions, and we assume that the observation includes the set of available actions. The agent then selects one action from the subset of the available variables. Formally, the space of actions is the set of all variables in all possible formulas in all solver states, where at every state only a small finite number of them is available. Practically, the agent will never see the effect of even a small part of these actions, and so it must generalize to unseen actions. An *episode* is the result of the interaction of the agent with the environment. We consider an episode to be *complete*, if the solver reaches a terminating state in the last step. As there are arbitrarily long episodes, we want to abort them after some step limit (the *decision limit*) and consider these episodes as *incomplete*.

### 3.1 BASELINES

While there are no competing learning approaches yet, human researchers and engineers have tried many heuristics for selecting the next variable. VSIDS is the best known heuristic for the solver we consider. It has been a dominant heuristic for SAT and several CDCL-based QBF algorithms for over 20 years now Moskewicz et al. (2001b); Eén & Sörensson (2003); Biere et al. (2009); Lonsing & Biere (2010); Rabe & Seshia (2016). We therefore consider VSIDS as the main baseline. VSIDS

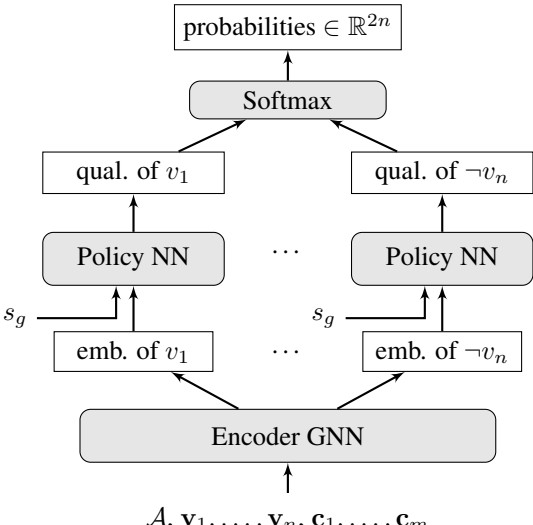

Figure 1: Sketch of the architecture for a formula $\varphi$ with $n$ variables $v_i$ and $m$ clauses. $s_g$ is the global state of the solver, $\mathcal{A}$ is the adjacency matrix, and $\mathbf{v}_i$ and $\mathbf{c}_i$ are the variable and clause labels.

maintains an *activity score* per variable and always chooses the variable with the highest activity that is still available. The activity reflects how often a variable recently occurred in conflict analysis. To select a literal of the chosen variable, VSIDS uses the Jeroslow-Wang heuristic (Jeroslow & Wang, 1990), which selects the polarity of the variable that occurs more often, weighted by the size of clauses they occur in. For reference, we also consider the *Random* heuristic, which chooses one of the available actions uniformly at random.

## 4 THE NEURAL NETWORK ARCHITECTURE

Our model gets an observation, consisting of a formula and the state of the solver, and selects one of the formula's literals (= a variable and a Boolean value) as its action. The model has two components: An *encoder* that produces an embedding for every literal, and a *policy network* that that rates the quality of each literal based on its embedding. We give an overview of the architecture in Fig. 1, describe the GNN in Subsection 4.1 and the policy network in Subsection 4.2.

### 4.1 A GNN ENCODER FOR BOOLEAN FORMULAS

In order to employ GNNs, we view the formula as a graph, where each clause and each literal is a node (see Fig. 2. For each literal in each clause, we draw an edge between their nodes. The resulting graph is bipartite and hence, we represent its edges as an $2n \times m$ adjacency matrix $\mathcal{A}$ with values in $\{0, 1\}$, where $2n$ is the number of literals and $m$ is the number of clauses. This graph structure determines the semantics of the formula except for the quantification of variables (i.e. whether a variable is universally or existentially quantified), which are provided as labels to the variables. For each variable $v$, the variable label $\mathbf{v} \in \mathbb{R}^{\lambda_V}$, with $\lambda_V = 7$, indicates whether the variable is universally or existentially quantified, whether it currently has a value assigned, and whether it was selected as a decision variable already on the current search branch. We use the variable label for both of its literals and by $\mathbf{v}_l$ we denote the label of the variable of $l$. For each clause $c$, the clause label $\mathbf{c} \in \mathbb{R}$ is a single scalar (in $\{0, 1\}$), indicating whether the clause was original or derived during conflict analysis.

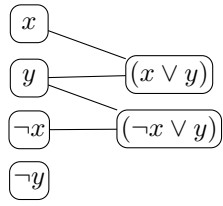

Figure 2: The bipartite graph for $(x \vee y) \wedge (\neg x \vee y)$.

While we are ultimately only interested in embeddings for literals, our GNN also computes embeddings for clauses as intermediate values. Literal embeddings have dimension $\delta_L = 16$ and clause embeddings have dimension $\delta_C = 64$. The GNN computes the embeddings over $\tau$ rounds. We define

the initial literal embedding as $l_0 = \mathbf{0}$, and for each round $1 \leq t \leq \tau$, we define the literal embedding $l_t \in \mathbb{R}^{\delta_L}$ for every literal $l$ and the clause embedding $c_t \in \mathbb{R}^{\delta_C}$ for every clause $c \in C$ as follows:

$$c_t = \text{ReLU}\big(\textstyle\sum_{l \in c} \mathbf{W}_L [\mathbf{v}_l^\top, l_{t-1}^\top, \bar{l}_{t-1}^\top] + \mathbf{B}_L\big), \quad \text{and } l_t = \text{ReLU}\Big(\textstyle\sum_{c, l \in c} \mathbf{W}_C [\mathbf{c}^\top, c_t^\top] + \mathbf{B}_C\Big).$$

The trainable parameters of our model are indicated as bold capital letters. They consist of the matrix $\mathbf{W}_L$ of shape $(2\delta_L + \lambda_V, \delta_C)$, the vector $\mathbf{B}_L$ of dimension $\delta_C$, the matrix $\mathbf{W}_C$ of shape $(\delta_C + \lambda_C, \delta_L)$, and the vector $\mathbf{B}_C$ of dimension $\delta_L$.

*Invariance properties.* The meaning of a formula in CNF is invariant under permutations of its clauses and of literals within each clause due to the commutativity of conjunction and disjunction. Our GNN architecture is invariant under these reorderings, as both conjunctions and disjunctions are computed through commutative operations (a sum), and, therefore, it cannot accidentally overspecialize to the ordering of clauses or literals. Swapping the literals of a variable does not change the truth of the formula either, and our GNN architecture respects that as well. The only place in our architecture where we use the information of which literals belong to the same variable is in the input to $c_t$. Depending on which literal of a variable occurs in the clause we order its literal embeddings differently. Lastly, note that variables are completely nameless in our representation.

### 4.2 POLICY NETWORK

The policy network predicts the quality of each literal based on the literal embedding and the global solver state. The *global solver state* is a collection of $\lambda_G = 5$ values that include only the essential parts of solver state that are not associated with any particular variable or clause. We provide additional details in Appendix A. The policy network thus maps the *final literal embedding* $[\mathbf{v}_l^\top, l_\tau^\top, \bar{l}_\tau^\top]$ concatenated with the global solver state to a single numerical value indicating the *quality* of the literal. The policy network thus has $\lambda_V + 2\delta_L + \lambda_G$ inputs, which are followed by two fully-connected layers. The two hidden layers use the ReLU nonlinearity. We turn the predictions of the policy network into action probabilities by a masked softmax. We mask all "illegal" actions, effectively ignoring the embeddings of variables which are universal, or are assigned already.

Note that the policy network predicts a score for each literal *independently*. All information about the graph that is relevant to the policy network must hence flow through the literal embedding. Since we experimented with graph neural networks with few iterations this means that *the quality of each literal is decided locally*. The rationale behind this design is that it is simple and efficient.

## 5 EXPERIMENTS

We conducted several experiments to examine whether we can improve the heuristics of the logic solver CADET through our deep reinforcement learning approach. [2]

- Q1 Can we learn to predict good actions for a family of formulas?
- Q2 How does the policy trained on short episodes generalize to long episodes?
- Q3 How well does the learned policy generalize to formulas from a different family of formulas?
- Q4 Does the improvement in the policy outweigh the additional computational effort? That is, can we solve more formulas in less time with the learned policy?

### 5.1 DATA

In contrast to most other works in the area, we evaluate our approach over a benchmark that (1) has been generated by a third party before the conception of this paper, and (2) is challenging to state-of-the-art solvers in the area. We consider a set of formulas representing the search for reductions between collections of first-order formulas generated by Jordan & Kaiser (2013), which we call *Reductions* in the following. Reductions is interesting from the perspective of QBF solvers, as its formulas are often part of the QBF competition. It consists of 4608 formulas of varying sizes and with varying degrees of hardness. On average the formulas have 316 variables; the largest formulas in the set have over 1600 variables and 12000 clauses. We filtered out 2573 formulas that are solved

---

[2]We provide the code and data of our experiments at `https://github.com/lederg/learningqbf`.

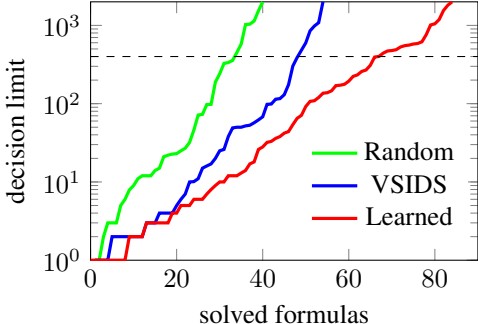 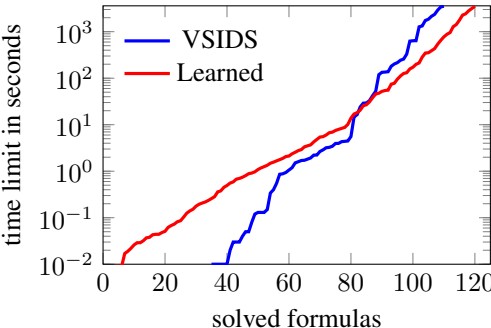

Figure 3: Two cactus plots showing how the number of solved formulas from the **test** set grows with increasing resource bounds. **Left:** Comparing the number of formulas solved with growing **decision limit** for Random, VSIDS, and our learned heuristic. **Right:** Comparing the number of formulas solved with growing **wall clock time**. Lower and further to the right is better.

without any heuristic decisions. In order to enable us to answer question 2 (see above), we further set aside a test set of 200 formulas, leaving us with a training set of 1835 formulas.

We additionally consider the 2QBF evaluation set of the annual competition of QBF solvers, QBFE-VAL (Pulina, 2016). This will help us to study cross-benchmark generalization.

## 5.2 REWARDS AND TRAINING

We jointly train the encoder network and the policy network using REINFORCE (Williams, 1992). For each batch we sample a formula from the training set, and generate $b$ episodes by solving it multiple times. In each episode we run CADET for up to 400 steps using the latest policy. Then we assign rewards to the episodes and estimate the gradient. We apply standard techniques to improve the training, including gradient clipping, normalization of rewards, and whitening of input data.

We assign a small negative reward of $-10^{-4}$ for each decision to encourage the heuristic to solve each formula in fewer steps. When a formula is solved successfully, we assign reward 1 to the last decision. In this way, we effectively treat unfinished episodes ($> 400$ steps) as if they take 10000 steps, punishing them strongly.

## 5.3 RESULTS

We trained the model described in Section 4 on the *Reductions* training set. We denote the resulting policy *Learned* and present the aggregate results in Figure 3 as a *cactus plot*, as usual for logic solvers. The cactus plot in Figure 3 indicates how the number of solved formulas grows for increasing decision limits on the *test set* of the *Reductions* formulas. In a cactus plot, we record one episode for each formula and each heuristic. We then sort the runs of each heuristic by the number of decisions taken in the episode and plot the series. When comparing heuristics, lower lines (or lines reaching further to the right) are thus better, as they indicate that more formulas were solved in less time.

We see that for a decision limit of 400 (dashed line in Fig. 3, left), i.e. the decision limit during training, Learned solved significantly more formulas than either of the baselines. The advantage of Learned over VSIDS is about as large as VSIDS over purely random choices. This is remarkable for the field and we can answer Q1 positively.

Figure 3 (left) also shows us that Learned performs well far beyond the decision limit of 400 steps that was used during its training. Observing the vertical distance between the lines of Learned and VSIDS, we can see that the advantage of Learned over VSIDS even grows exponentially with an increasing decision limit. (Note that the axis indicating the number of decisions is log-scaled.) We can thus answer Q2 positively.

A surprising fact is that small and shallow neural networks already achieved the best results. Our best model uses $\tau = 1$, which means that for judging the quality of each variable, it only looks at the variable itself and the immediate neighbors (i.e. those variables it occurs together with in a constraint).

The hyperparameters that resulted in the best model are $\delta_L = 16$, $\delta_C = 64$, and $\tau = 1$, leading to a model with merely 8353 parameters. The small size of our model was also helpful to achieve quick inference times.

To answer Q3, we evaluated the learned heuristic also on our second data set of formulas from the QBF solver competition QBFEVAL. Random solved 67 formulas, VSIDS solved 125 formulas, and Learned solved 111 formulas. The policy trained on *Reductions* significantly improved over random choices, but does not beat VSIDS. This is hardly surprising, as our learning approach specialized the solver to a specific—different—distribution of formulas. Also it must be taken into account that the solver CADET has been tuned to QBFEVAL over year, and hence may perform much stronger on QBFEVAL than on the Reductions benchmark. We include further cross-benchmark generalization studies in the Appendix.

To answer our last question, Q4, we compare the runtime of CADET in with our learned heuristic to CADET with the standard VSIDS heuristic. In Fig. 3 (right) we see that for small time limits (up to 10 seconds), VSIDS still solves more formulas than the learned heuristic. But, for higher time limits, the learned heuristic starts to outperform VSIDS. For a time limit of 1 hour, we solved 120 formulas with the learned heuristic while only 110 formulas were solved with VSIDS (see right top corner). Conversely, for solving 110 formulas the learned heuristic required a timeout of less than 12 minutes, while VSIDS took an hour. Furthermore, our learning and inference implementation is written in Python and not particularly optimized. The NN agent is running in a different process from CADET, and incurs an overhead per step for inter-process communication and context switches, which is enormous compared to the pure C implementation of CADET using VSIDS. This overhead could be easily reduced, and so we expect the advantage of our approach to grow.

# 6 RELATED WORK

Independent from our work, GNNs for Boolean logic have been explored in NeuroSAT (Selsam et al., 2018), where the authors use it to solve the SAT problem directly. While using a similar neural architecture, the network is not integrated in a state-of-the-art logic solver, and does not improve the state of the art in performance. Selsam & Bjørner (2019) recently extended NeuroSAT to use its predictions in a state-of-the-art SAT solver. In contrast to their work, we integrate GNNs much tigher into the solver and train the heuristics directly through reinforcement learning. Thus allow deep learning to take direct control of the solving process. Also, we focus on QBF instead of SAT, which strongly affects the runtime tradeoffs between spending time on "thinking" about a better decision versus executing many "stupid" decisions.

Amizadeh et al. (2019) suggest an architecture that solves circuit-SAT problems. Unlike NeuroSAT, and similar to our approach, they train their model directly to find a satisfying assignment by using a differentiable "soft" satisfiability score as their loss. However, like NeuroSAT, their approach aims to solve the problem from scratch, without leveraging an existing solver, and so is difficult to scale to state-of-the-art performance. They hence focus on small random problems. In contrast, our approach improves the performance of a state-of-the-art algorithm. Furthermore, our learned heuristic applies to SAT and UNSAT problems alike.

Yang et al. (2019) extended the NeuroSAT architecture to 2QBF problems. In contrast to our work, they do not embed their GNN model in a modern DPLL solver, and instead try to predict good counter-examples for a CEGAR solving approach. They focus on formulas with 18 variables, which are trivial for state-of-the-art solvers. Chen & Yang (2019) showed that a pure GNN approach is unable to solve Boolean formulas when they are unsatisfiable, which in our work is addressed by combining GNNs with a logic reasoning engine.

Reinforcement learning has been applied to other logic reasoning tasks. Kaliszyk et al. (2018) recently explored learning linear policies for tableaux-style theorem proving. Kurin et al. (2019) follow a similar approach to ours for SAT solvers, but only evaluate on small synthetic formulas and do not improve the overall performance of the underlying SAT solver. Kusumoto et al. (2018) applied reinforcement learning to propositional logic in a setting similar to ours; just that we employ the learning in existing strong solving algorithms, leading to much better scalability. Balunovic et al. (2018) use deep reinforcement learning to improve the application of *high-level* strategies in SMT solvers, but do not investigate a tighter integration of deep learning with logic solvers. Also other

works on combinatorial search explored the use of GNNs (some trained with reinforcement learning) for problems such as random SAT (Yolcu & Póczos, 2019), coloring graphs (Huang et al., 2019), and MILP (Gasse et al., 2019).

Most previous approaches that applied neural networks to logical formulas used LSTMs or tree models syntax-tree of formulas (Bowman et al., 2014; Irving et al., 2016; Allamanis et al., 2016; Loos et al., 2017; Evans et al., 2018; Chvalovsky, 2019; Chen & Tian, 2018) or classical ML models (Gauthier et al., 2017; Kaliszyk et al., 2018; Soos et al., 2019). Instead, we suggest a GNN approach, based on a graph-view on formulas in CNF. Recent work suggests that GNNs appear to be a good architecture for logics (Paliwal et al., 2019; Wang et al., 2017). Bansal et al. (2019); Huang et al. (2018); Yang & Deng (2019) provide a learning environments around interactive theorem provers.

Other competitive QBF algorithms include expansion-based algorithms (Biere, 2004; Pigorsch & Scholl, 2010), CEGAR-based algorithms (Janota & Marques-Silva, 2011; 2015; Rabe & Tentrup, 2015), circuit-based algorithms (Klieber, 2012; Tentrup, 2016; Janota, 2018a;b), and hybrids (Janota et al., 2012; Tentrup, 2017). Recently, Janota (2018a) successfully explored the use of (classical) machine learning techniques to address the generalization problem in QBF solvers.

## 7 CONCLUSIONS

We presented an approach to improve the heuristics of a backtracking search algorithm for Boolean logic through deep reinforcement learning. Our approach brings together the best of two worlds: The superior flexibility and performance of intuitive reasoning of neural networks, and the ability to explain (prove) results in formal reasoning. The setting is new and challenging to reinforcement learning; QBF is a very general, combinatorial problem class, featuring an unbounded input-size and action space. We demonstrate that these problems can be overcome, and reduce the overall execution time of a competitive QBF solver by a factor of 10 after training on similar formulas.

This work demonstrates the huge potential that lies in the tight integration of deep learning and logical reasoning algorithms, and hence motivates more aggressive research efforts in the area. Our experiments suggest two challenges that we want to highlight: (1) We used very small neural networks, and—counterintuitively—larger neural networks were not able to improve over the small ones in our experiments. (2) The performance overhead due to the use of neural networks is large; however we think that with more engineering effort we could be significantly reduce this overhead.

**Acknowledgements.** This work was supported in part by National Science Foundation (NSF) grants CNS-1836601, CNS-1446619, CNS-1739816, and CCF-1837132, by the iCyPhy center, and by Berkeley Deep Drive. The second author was affiliated with UC Berkeley during the initial part of this work.

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

## A   GLOBAL SOLVER STATE

1. Current decision level
2. Number of restarts
3. Restarts since last major restart
4. Conflicts until next restart
5. Ratio of variables that already have a Skolem function to total variables. Formula is solved when this reaches 1.

## B   LITERAL LABELS

Here we describe the details of the variable labels presented to the neural network described in Section 4. The vector $\mathbf{v}$ consists of the following 7 values:

$$
\begin{aligned}
y_0 \in \{0,1\} \quad & \text{indicates whether the variable} \\
& \text{is universally quantified,} \\
y_1 \in \{0,1\} \quad & \text{indicates whether the variable} \\
& \text{is existentially quantified,} \\
y_2 \in \{0,1\} \quad & \text{indicates whether the variable} \\
& \text{has a Skolem function already,} \\
y_3 \in \{0,1\} \quad & \text{indicates whether the variable} \\
& \text{was assigned constant True,} \\
y_4 \in \{0,1\} \quad & \text{indicates whether the variable} \\
& \text{was assigned constant False,} \\
y_5 \in \{0,1\} \quad & \text{indicates whether the variable} \\
& \text{was decided positive,} \\
y_6 \in \{0,1\} \quad & \text{indicates whether the variable} \\
& \text{was decided negative, and}
\end{aligned}
$$

## C   THE QDIMACS FILE FORMAT

QDIMACS is the standard representation of quantified Boolean formulas in prenex CNF. It consists of a header "`p cnf <num_variables> <num_clauses>`" describing the number of variables and the number of clauses in the formula. The lines following the header indicate the quantifiers. Lines starting with 'a' introduce universally quantified variables and lines starting with 'e' introduce existentially quantified variables. All lines except the header are terminated with 0; hence there cannot be a variable named 0. Every line after the quantifiers describes a single clause (i.e. a disjunction over variables and negated variables). Variables are indicated simply by an index; negated variables are indicated by a negative index. Below give the QDIMACS representation of the formula $\forall x. \exists y. \ (x \vee y) \wedge (\neg x \vee y)$:

```
p cnf 2 2
a 1 0
e 2 0
1 2 0
−1 2 0
```

There is no way to assign variables strings as names. The reasoning behind this decision is that this format is only meant to be used for the computational backend.

## D   HYPERPARAMETERS AND TRAINING DETAILS

We trained a model on the reduction problems training set for 10M steps on an AWS server of type C5. We trained with the following hyperparameters, yet we note that training does not seem overly sensitive:

- Literal embedding dimension: $\delta_L = 16$

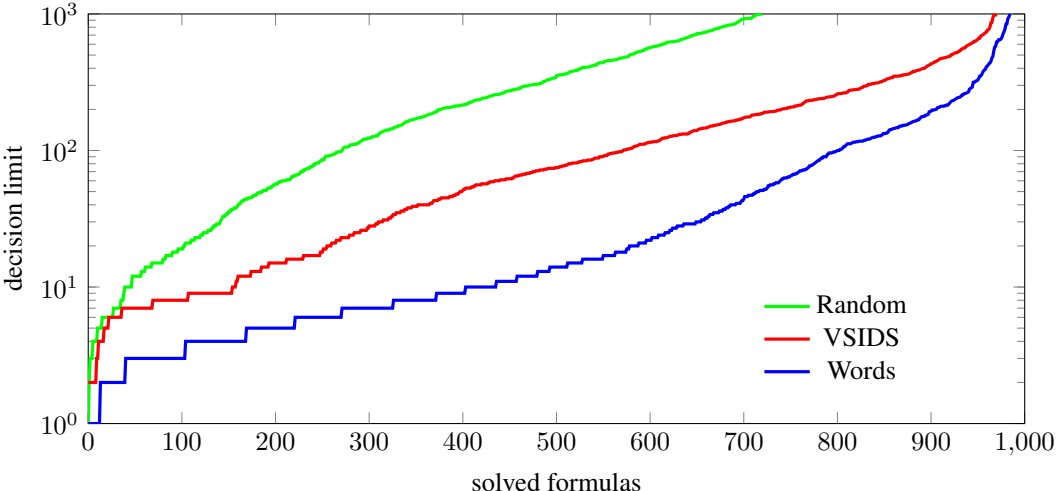

Figure 4: A cactus plot describing how many formulas from the **test** set were solved within growing decision limits on the *Words* test set. Lower and further to the right is better.

- Clause embedding dimension: $\delta_C = 64$
- Learning rate: 0.0006 for the first 2m steps, then 0.0001
- Discount factor: $\gamma = 0.99$
- Gradient clipping: 2
- Number of iterations (size of graph convolution): 1
- Minimal number of timesteps per batch: 1200

## E    ADDITIONAL DATASETS AND EXPERIMENTS

While the set of Reductions-formulas we considered in the main part of the paper was created independently from this paper and is therefore unlikely to be biased towards our approach, one may ask if it is just a coincidence that our approach was able to learn a good heuristic for that particular set of formulas. In this appendix we consider two additional sets of formulas that we call *Boolean* and *Words*, and replicated the results from the main part. We show that we can learn a heuristic for a given set/distribution of formulas that outperforms VSIDS by a significant margin.

*Boolean* is a set of formulas of random circuits. Starting from a fixed number (8) of Boolean inputs to the circuit, individual AND-gates are added (with randomly chosen inputs with random polarity) up to a certain randomized limit. This circuit is turned into a propositional Boolean formula using the Tseitin transformation, and then a small fraction of random clauses is added to add some irregularities to the circuit. (Up to this point, the process is performed by the fuzz-tester for SAT solvers, FuzzSAT, available here http://fmv.jku.at/fuzzsat/.) To turn this kind of propositional formulas into QBFs, we randomly selected 4 variables to be universally quantified. This resulted in a more or less even split of true and false formulas. The formulas have 50.7 variables on average. In Figure 5 we see that training a model on these formulas (we call this model *Boolean*, like the data set) results in significantly better performance than VSIDS. The advantage of the learned heuristic over VSIDS and Random is smaller compared to the experiments on Reductions in the main part of the paper. We conjecture that this is due to the fact that these formulas are much easier to begin with, which means that there is not as much potential for improvement.

*Words* is a data set of random expressions over (signed) bitvectors. The top-level operator is a comparison ($=, \leq, \geq, <, >$), and the two subexpressions of the comparison are arithmetic expressions. The number of operators and leafs in each expression is 9, and all bitvectors have word size 8. The expressions contain up to four bitvector variables, alternatingly assigned to be existentially and

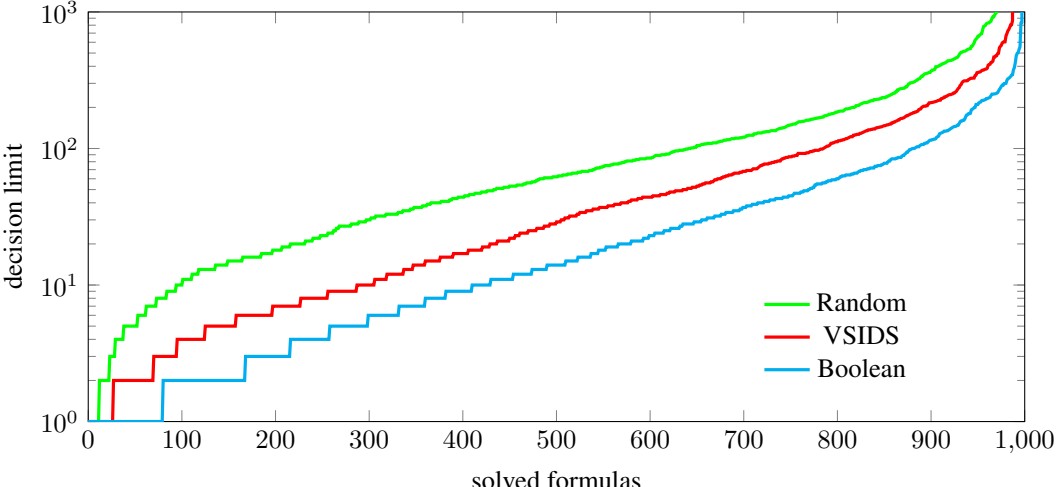

Figure 5: A cactus plot describing how many formulas from the **test** set were solved within growing decision limits on the *Boolean* test set. Lower and further to the right is better.

universally quantified. The formulas are simplified using the circuit synthesis tool ABC, and then they are turned into CNF using the standard Tseitin transformation. The resulting formulas have 71.4 variables on average and are significantly harder for both Random and VSIDS. For example, the first formula from the data set looks as follows: $\forall z.\exists x.((x-z) \text{ xor } z) \neq z+1$, which results in a QBF with 115 variables and 298 clauses. This statement happens to be true and is solved with just 9 decisions using the VSIDS heuristic. In Figure 4 we see that training a new model on the *Words* dataset again results in significantly improved performance. (We named the model *Words*, after the data set.)

We did not include the formula sets Boolean and Words in the main part, as they are generated by a random process - in contrast to Reductions, which is generated with a concrete application in mind. In the formal methods community, artificially generated sets of formulas are known to differ from application formulas in non-obvious ways.

## F  ADDITIONAL EXPERIMENTS ON GENERALIZATION TO LARGER FORMULAS

An interesting observation that we made is that models trained on sets of small formulas generalize well to larger formulas from similar distributions. To demonstrate this, we generated a set of larger formulas, similar to the *Words* dataset. We call the new dataset *Words30*, and the only difference to *Words* is that the expressions have size 30. The resulting formulas have 186.6 variables on average. This time, instead of training a new model, we test the model trained on *Words* (from Figure 4) on this new dataset.

In Figure 6, we see that the overall hardness (measured in the number of decisions needed to solve the formulas) has increased a lot, but the relative performance of the heuristics is still very similar. This shows that the heuristic learned on small formulas generalizes relatively well to much larger/harder formulas.

In Fig. 3, we have already observed that the heuristic also generalizes well to much longer episodes than those it was trained on. We believe that this is due to the "locality" of the decisions we force the network to take: The graph neural network approach uses just one iteration, such that we force the heuristics to take very local decisions. Not being able to optimize globally, the heuristics have to learn local features that are helpful to solve a problem sooner rather than later. It seems plausible that this behavior generalizes well to larger formulas (Fig. 6) or much longer episodes (Fig. 3).

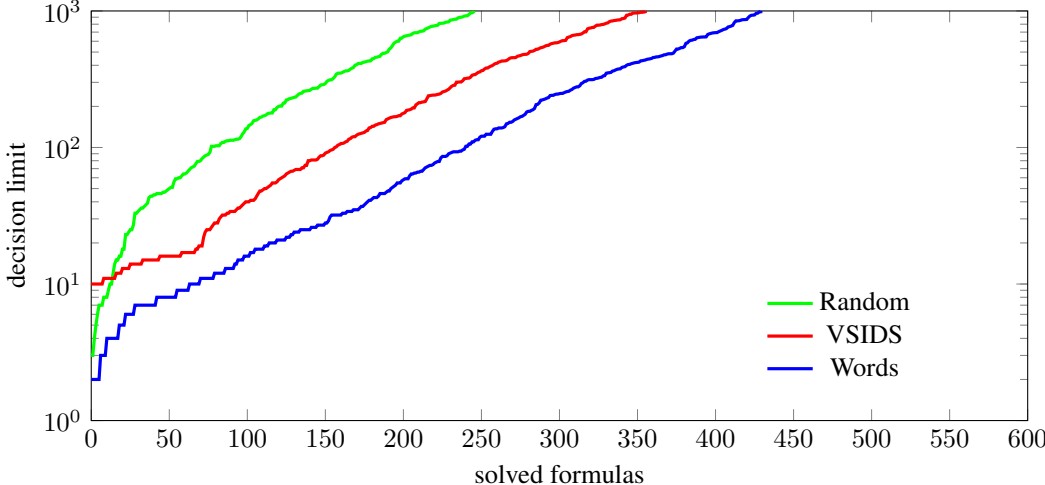

Figure 6: A cactus plot describing how many formulas were solved within growing decision limits on the *Words30* test set. Lower and further to the right is better. Note that unlike in the other plots, the model *Words* was not trained on this distribution of formulas, but on the same *Words* dataset as before.

## G    ENCODER VARIANTS AND HYPERPARAMETERS

The encoder described in Subsection 4.1 is by no means the only reasonable choice. In fact, the graph representation described in Fig. 2 is not unique. One could just as well represent the formula as a bipartite graph on *variables* and clauses, with two types of edges, one for each polarity. The encoder then produces encodings of variables rather than literals, and the propagation along edges is performed with two different learned parameter matrices, one for each edge type. The equations for such an encoder are:

$$c_t = \text{ReLU}\left(\sum_{l \in c} \mathbf{W}_V[\mathbf{v}^\top, v_{t-1}^\top] + \mathbf{B}_V\right)$$

$$v_t = \text{ReLU}\left(\sum_{c, v \in c} \mathbf{W}_C[\mathbf{c}^\top, c_t^\top] + \mathbf{B}_C\right)$$

Where $W_V$ is one of $W_V^+, W_V^-$ (and similarly, $W_C \in \{W_C^+, W_C^-\}$), depending on the polarity of the occurence of $v$ in $c$, with $\mathbf{v}$ as the variable's label. Accordingly, we change the policy network to produce *two* scores per variable embedding $v_\tau$, as the qualities of assigning this variable to positive or negative polarity. In our experiments, this variant of the encoder achieved comparable results to those of the literal-based encoder.

The hyperparameter $\tau$ controls the number of iterations within the GNN. Here too, there are several variants of the encoder one could consider. The architecture described in Subsection 4.1, which achieved the reported results, applies the same transformation for every iteration (the matrices $W_C$, $W_L$). We've also experimented with a variant that uses $\tau$ different learned transformations, one per iteration, denoted $W_C^t, W_L^t$, for $1 \le t \le \tau$ (intuitively, this allows the network to perform a different computation in every iteration). It achieved comparable results, yet with roughly $\tau$ times the number of parameters. A version with even more parameters gave the $t'th$ transformation access not only to the $t - 1$ embedding, but to all the $1, \ldots, t - 1$ previous embeddings, through residual connections. This version also didn't achieve significantly better results. To get results with more than one iteration we had to add a normalization layer between every two iterations. We experimented with both Layer Normalization (Ba et al., 2016) and a GRU cell (Chung et al., 2014), which gave

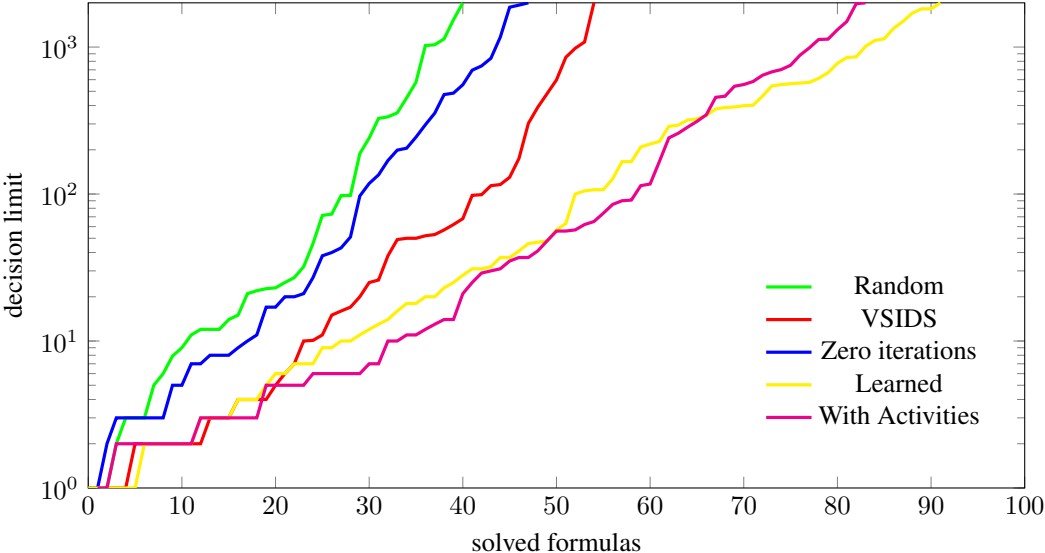

Figure 7: A cactus plot describing how many formulas were solved within growing decision limits on the reduction test set with different models. VSIDS, Random, and Learned are same as left side of Figure 3.

similar results. Adding a 2nd and 3rd iteration achieved only slightly better results when measuring number of decisions to solve a formula, at the cost of more parameters, slower training, and more importantly, slower inference at runtime. When measuring number of formulas solved in real time, a single iteration achieved best results overall. However, given the large overhead of our agent implementation, it is possible that an optimized in-process implementation could still benefit from multiple iterations in the GNN.

It is interesting to point out that when we tested a model with *zero* iterations, that is, no GNN at all, where the policy network gets to see only the variable labels from the solver, it achieved results that were better than Random and clearly demonstrated learning, but worse than VSIDS, and considerably less than the results for 1 iteration. That shows that at least the 1-hop neighborhood of a variable contains information which is crucial, we cannot achieve comparable results without considering this local topology of the graph.

Another interesting observation is that the model which achieved best results did not have access to the variable VSIDS activity scores! Adding activity scores to the variable feature vectors in fact slightly degraded performance. It learns *faster*, but converges to a lower average reward, and performs slighly worse on the validation and test sets, especially on the harder problems. We hypothesize that this is because the model learns to rely on the activity scores, and they will be quite different in harder (longer) episodes, and outside the range it trained on. Furthermore, it shows that it is possible to achieve a heuristic which performs better than VSIDS without even computing activities!

The results for the different variants can be seen in Figure 7.

