# OpenReview forum: "Learning Heuristics for Quantified Boolean Formulas through Reinforcement Learning"
_ICLR.cc/2020/Conference — Accept (Poster)_

### Official Review · AnonReviewer3 · 2019-10-20
**Official Blind Review #3**

**Rating:** 3

**Review:**

This paper investigates the problem of predicting the truth of quantified boolean formulae using deep reinforcement learning. In this setting, the problem is formulated as a reinforcement learning task, in which the learner is interacting with a solver (CADET), and its goal is to find a sequence of actions (each associated with a choice of a variable and a value) in order to reach a terminal state as fast as possible. The neural architecture includes a GNN encoder for the input formula, a policy neural net for iteratively assessing the quality of literals, and a final softmax layer for choosing the final literal. Experiments, performed on various 2QBF instances, address several questions such as the ability to compete with existing heuristics (VSIDS) in CADET and to generalize predictions on long episodes or different formulae.

Overall, the paper is well-motivated. The introduction is well-written and explains the interest of learning new heuristics for QBF problems. The learning framework is relatively simple and elegant. Unfortunately, the paper suffers from many clarity issues in the problem formulation, the neural-net architecture, and the experiments. So, it is quite difficult to accept the paper in its current state.

Section 2: In this section, some background knowledge about boolean problems and solvers are provided. Although the second paragraph about CNF formulae and CDCL solvers is well-written, the third paragraph about QBF should be clarified. For QBF formulae, the authors write “The algorithmic problem considered for QBF is to determine the truth of a quantified formula (TQBF)”. Well, this is not an algorithmic problem, but a decision problem. Furthermore, what is TQBF? I guess that the authors are talking about 2-QBF, for which the prenex is of the form $\forall \exists$. Here, the decision task should be explained in more detail using, for example, a game tree for explaining the distinct roles of “for all” variables and “there is” variables. As the decision problem for 2-QBF is more complex than the satisfiability problem for CNF, this point should be emphasized (i.e. the decision problem for 2-QBF is $\Pi_2^P$ complete).

Section 3: The problem of predicting the truth of 2-QBF is formulated as an MDP, where the environment is essentially controlling the input instances (of 2-QBF) and the states of the solver, and the learner’s actions are variable-value selections. First of all, I am not entirely convinced that an MDP is the right choice for specifying this problem. Basically, 2-QBF is a two-player game (“for all” vs “there is”), and the goal of the solver is to play “there is” by finding a satisfying assignment for each possible play (i.e. variable assignment) of the “for all” player. So, a natural framework here would be a stochastic game (SG) which generalizes the MDP framework. Actually, in the present MDP framework there are no distinctions between “for all” variables and “there is” variables. It seems that the learner can choose “for all” variables (which is wrong). Furthermore, the MDP is ill-defined. It is said that a policy is a mapping $\pi: S \times A$. This is ill-defined: what is the range of $\pi$? Usually, a (mixed) policy is a mapping $\pi$ from $S$ into the $|A|$-dimensional simplex. The reward function is also ambiguous: it is using a discount factor $\gamma$ but, unless I missed something, this factor is not clarified in the rest of the paper. Finally, what is an “episode”? In the paper it is said “An episode is the result of the interaction of the agent with the environment”. Well, this is quite unclear. Usually, in an episodic-MDP the state space is partitioned into layers, i.e. $X = \bigcup_{i = 0}^L X_i$, where $X_0$ is a singleton set (the initial state), and $X_L$ specifies the terminal states. Transitions are possible only between consecutive layers.  According to this usual framework, an episode is a sequence of actions made by the agent, starting from $X_0$ and moving forward across the consecutive layers until it reaches a state in $X_L$. For the 2-QBF decision problem, each terminal state in $X_L$ would naturally consists in the affectation of each variable in the prenex to a Boolean value. But this is not clear in the paper, because the authors are saying that “We consider an episode to be complete, if the solver reaches a terminal state in the last step”. This would mean that the number of actions per episodes is capped, and hence, the agent can reach a terminal, yet non-final, decision state.

Section 4: The overall architecture (GNN encoder + Policy Network) is relatively standard, but the choice of the constants for dimension parameters $\lambda_V$, $\delta_L$ and $\delta_C$ is a bit disconcerting. Notably, $\lambda_V$ is used to capture the features of variables. Specifically, it is written that “$\lambda_V = 7$ indicates whether the variable is universally or existentially quantified, whether it currently has a value assigned and whether it was selected as a decision variable already on the current search branch.” But what is the difference between the second feature and the third one? If a variable has already been branched, then it has been assigned, and conversely. Furthermore, if you have 3 boolean features $\lambda_V$ should be fixed to 3. So, why choosing 7? For $\delta_L$ and $\delta_C$, it seems that their values correspond to the “best model”. But how this model is chosen? Did the author perform some grid search to find those values?  Finally, some comments about the last layer of the architecture (softmax function) would be welcome. In the end, we get a probability distribution over literals (agent’s available actions) “after masking illegal actions” (as written by the authors). But what is an illegal action? Is it a literal defined on a universally quantified variable? A literal defined on an already assigned existential variable?

Section 5: In the experiments, the authors are examining four different questions, which are all interesting. But the experimental setup and the reported results are quite unclear. In fact, the experimental setup looks wrong, because if the “Reductions” dataset is taken from Jordan & Kaiser (SAT’13), it consists of formulae for which the prenex is of the form $\exists \forall$”. Unless I am wrong, this is the inverse of the 2QBF problem examined in the present paper  - Jordan and Kaiser were examining a $\Sigma_2^P$-complete problem, while you are examining a $\Pi_2^P$-complete problem. So, did you reverse the quantifiers for making experiments? This should be clarified in the paper. Furthermore, for this dataset which originally consists of 4500 instances, the authors say that “We filtered out 2500 formulas that are solved without any heuristic decisions.” What does this mean? Are all those 2500 formulas containing only universal quantifiers? In the remaining 2000 instances, what is the ratio between universally quantified variables and existentially quantified ones? By the way, how can we get 1835 training instances? 4500 - 2500 - 200 = 1800.

In the experimental results, which formulae have been used to compute the cactus plots? Training instances (1800)? Test instances (200)? Both of them? For training instances, the protocol reported in Section 5.2 is relatively clear. But for test instances, what is the protocol? Is CADET using the best policy trained on the 1800 instances for solving the remaining 200 instances? Furthermore, the notion of “decision limit” is quite confusing. According to Section 5.2, it seems that the decision limit is the horizon of each episode, i.e. the number of calls to the solver CADET using the latest policy estimated by the NN architecture. But this should be clarified unambiguously.


**Experience Assessment:**

I have read many papers in this area.

**Review Assessment: Checking Correctness Of Derivations And Theory:**

I assessed the sensibility of the derivations and theory.

**Review Assessment: Checking Correctness Of Experiments:**

I assessed the sensibility of the experiments.

**Review Assessment: Thoroughness In Paper Reading:**

I read the paper thoroughly.

---

> ### Author Response · Authors · 2019-11-14
> **Thank you for your detailed comments.**
>
> We believe that review #3 contains some misunderstandings, which we address first:
>
> >This paper investigates the problem of predicting the truth of quantified boolean
> >formulae using deep reinforcement learning.
>
> This paper does not predict the truth of formulas, but instead predicts heuristic decisions in a logical reasoning algorithm.
>
> >I am not entirely convinced that an MDP is the right choice for specifying
> >this problem.
>
> The presentation of the logical reasoning environment as an MDP is required because we use reinforcement learning. We will elaborate this point in the final version.
>
> >Actually, in the present MDP framework there are no distinctions between
> >“for all” variables and “there is” variables.
>
> Variables are labeled with their quantifier as described in Section 4 and again in Appendix B.
>
> >But what is the difference between the second feature and the third one? If a
> >variable has already been branched, then it has been assigned, and conversely.
>
> This assumption is incorrect. Variables can also be assigned through propagation.
>
> >For the 2-QBF decision problem, each terminal state in X_L would naturally
> >consists in the affectation of each variable in the prenex to a Boolean value.
>
> The reasoning algorithm on which we build follows follows a very different approach as we explain in Section 2.
>
>
>
> In the following we address the reviewer’s questions:
>
> >“The algorithmic problem considered for QBF is to determine the truth of a quantified formula
> >(TQBF)”. Well, this is not an algorithmic problem, but a decision problem. Furthermore, what
> >is TQBF? I guess that the authors are talking about 2-QBF, for which the prenex is of the
> >form [AE].
>
> TQBF is the problem to determine the truth of a given quantified Boolean formula. And, indeed, this work focuses on 2QBF in prenex CNF, as explained in Section 2. We now emphasize 2QBF in the abstract and introduction.
>
> >Here, the decision task should be explained in more detail using, for example, a game tree
> >for explaining the distinct roles of “for all” variables and “there is” variables.
>
> The reasoning algorithm underlying this work does not follow the game semantics of QBF.
>
> >what is the range of [the policy] pi?
>
> It is the interval [0,1]. We fixed it.
>
> >what is an “episode”?
>
> We roll out the interaction between the policy and the solver up to a given number of rounds (the decision limit). If the prover solves the formula before the decision limit the episode may be shorter than the decision limit.
>
> >[...] if you have 3 boolean features lambda_V should be fixed to 3. So, why choosing 7?
>
> Please check out Appendix B for a detailed description of the variable labels. We chose a natural and typical, yet somewhat verbose, encoding.
>
> >But what is an illegal action? Is it a literal defined on a universally quantified
> >variable? A literal defined on an already assigned existential variable?
>
> Both. We clarified this in the updated version.
>
> >But how this model is chosen? Did the author perform some grid search to find
> >those values?
>
> We did not have the chance to do a full grid search, but experimented with various settings.
>
> >So, did you reverse the quantifiers for making experiments?
>
> No. We took the formulas from qbflib.org as is, without modifying them whatsoever. Reversing the quantifiers would fundamentally change the meaning of the formulas.
>
> >Furthermore, for this dataset which originally consists of 4500 instances, the
> >authors say that “We filtered out 2500 formulas that are solved without any heuristic
> >decisions.” What does this mean? Are all those 2500 formulas containing only universal
> >quantifiers?
>
> No, these formulas were solved by unit propagation, propagation of unique Skolem functions, and pure literal analysis alone. These techniques are part of the core of CADET and do not involve any heuristic decisions.
>
> >By the way, how can we get 1835 training instances? 4500 - 2500 - 200 = 1800
>
> The exact numbers are 4608 formulas overall, 2573 filtered out as they were solved without heuristic decisions, 200 withheld for testing, 1835 remaining formulas to train on.
>
> >In the experimental results, which formulae have been used to compute the cactus plots?
>
> The test formulas. We augmented the captions.
>
> >Is CADET using the best policy trained on the 1800 instances for solving the
> >remaining 200 [test] instances?
>
> Yes.
>
> >Furthermore, the notion of “decision limit” is quite confusing. According to Section 5.2,
> >it seems that the decision limit is the horizon of each episode, i.e. the number of calls to
> >the solver CADET using the latest policy estimated by the NN architecture. But this should
> >be clarified unambiguously.
>
> Yes, the decision limit is the horizon of each episode. We clarified this in the updated version.
>
> Minor corrections omitted due to character limit.

---

> > ### Author Response · Authors · 2019-11-14
> > **Additional comment on the use of masked softmax**
> >
> > >Finally, some comments about the last layer of the architecture (softmax function)
> > >would be welcome.
> >
> > As explained, the "allowed" actions are the existential variables which are not yet assigned, and so we use a masked softmax, effectively ignoring the predictions for illegal actions. It does indeed feel a bit unsatisfactory that the network assigns probability mass to illegal actions, because we would have liked the network to "learn" at the very least that it should not do something silly like choosing a universal variable. We've experimented around this point, for example by allowing illegal actions but then aborting the episode with a large penalty, or masking them but also adding an auxiliary loss to penalize the probability mass under the illegal actions (which is indeed equivalent to a simple version of the recently introduced "semantic loss" concept). These variations achieved comparable results to straightforward masking, and so we omitted them for simplicity.

---

### Official Review · AnonReviewer2 · 2019-10-20
**Official Blind Review #2**

**Rating:** 8

**Review:**

This paper proposed a GNN to improve an existing search based solver for 2-QBF solvers. The neural network is being used to predict the next steps in the search procedure, in this case, assignment to literals of the 2-QBF formula. Justifiably, a reinforcement learning formulation is used. I thought that the paper was very impressive. All necessary concepts were clearly introduced, the claims were very clear and thoroughly validated. Limitations of the current approach are also properly discussed.

Only comment I have is that using shallow networks with one iteration sounds not enough for a problem like 2-QBF solving. I noticed that you have this exploration in the appendix, would recommend moving it to the main paper. I would also have liked 2-QBF to be mentioned more explicitly in the abstract and early introduction.

Minor errors:
"and" --> "an" in Sec 1
"variale" --> "variable" in Sec 4.2
Reference ? in Sec 6

**Experience Assessment:**

I do not know much about this area.

**Review Assessment: Checking Correctness Of Derivations And Theory:**

I assessed the sensibility of the derivations and theory.

**Review Assessment: Checking Correctness Of Experiments:**

I carefully checked the experiments.

**Review Assessment: Thoroughness In Paper Reading:**

I read the paper thoroughly.

---

> ### Author Response · Authors · 2019-11-08
> **We thank reviewer #2 for their time to review this paper.**
>
> > I would also have liked 2-QBF to be mentioned more explicitly in the abstract and
> > early introduction.
>
> We updated the paper to clearly mention 2QBF in the abstract and introduction. We also fixed the minor errors pointed out in the review.
>
> > Only comment I have is that using shallow networks with one iteration sounds not enough
> > for a problem like 2-QBF solving. I noticed that you have this exploration in the
> > appendix, would recommend moving it to the main paper.
>
> We fully agree that, intuitively, deeper networks should be better at the task, and we were surprised that the performance-quality tradeoff turned out as it is. We do not think that the negative result on networks with additional iterations adds actionable insights to the readers and therefore moved it to the appendix. It is available on arXiv for interested readers.

---

### Official Review · AnonReviewer1 · 2019-10-23
**Official Blind Review #1**

**Rating:** 6

**Review:**


This will be an uncharacteristically short review. The work poses an interesting idea: why not mix heuristics and learning. It reads as if the paper was written a while ago and the intro was not updated, since there is a lot of related work using the same concept. Please cite existing work in the introduction, it reflects negatively on the paper.

The elephant in the room is that I have read this paper before. The work has a number of citations and has sparked a good amount of follow-up work. I like the ideas and they have received enough scrutiny already. It was not the best decision for ICLR 2019 to have rejected this paper, honestly. I will argue to accept the paper if the references are updated, it deserves a wider audience.

That said, I don't like this GNN embedding of the QBF. The negation should have been defined as an edge attribute, not through nodes (as in (Yolcu and Póczos, 2019)). The representation also does not encode the quantifiers well but I feel this is a question for future work.

Minor comments:
- Please update your paper. It needs a good refresh with the recent literature that cites your work.

- "An intriguing question for artificial intelligence is: can (deep) learning be effectively used for symbolic reasoning?" => Can representation learning be effectively used for symbolic reasoning? This is one of the most intriguing question in artificial intelligence today.



**Experience Assessment:**

I have read many papers in this area.

**Review Assessment: Checking Correctness Of Derivations And Theory:**

I assessed the sensibility of the derivations and theory.

**Review Assessment: Checking Correctness Of Experiments:**

I assessed the sensibility of the experiments.

**Review Assessment: Thoroughness In Paper Reading:**

I read the paper at least twice and used my best judgement in assessing the paper.

---

> ### Author Response · Authors · 2019-11-08
> **We thank reviewer #1 for their time to review this paper**
>
> We updated the references. The area is growing quickly and it is easy to miss new papers. Please let us know if there are papers that we did not cite.
>
> > I don't like this GNN embedding of the QBF. The negation should have been defined as an edge attribute, not through nodes (as in (Yolcu and Póczos, 2019)).
>
> We agree that the encoding of negations in edge types as in Yolcu and Póczos 2019 is an elegant alternative to our encoding. Note that our design only requires a single edge type. We have experimented with alternatives with multiple edge types but did not find a clear performance advantage of these designs.
>
> > The representation also does not encode the quantifiers well but I feel this is a question for future work.
>
> Our approach to encode quantifiers is tailored to 2QBF. We avoided more explicit ways to encode quantifiers as we were concerned about efficiency.

---

### Decision · Program_Chairs · 2019-12-19

**Decision:**

Accept (Poster)

**Comment:**

This paper proposes a new method to learning heuristics for quantified boolean formulas through RL. The focus is on a method called backtracking search algorithm. The paper proposes a new representation of formulas to scale the predictions of this method.

The reviewers have an overall positive response to this paper. R1 and R2 both agree that the paper should be accepted, and have given some minor feedback to improve the paper. R3 initially was critical of the paper, but the rebuttal helped to clarify their doubt. They still have one more comment and I encourage the authors to address this in the final version of the paper.

R3 meant to increase their score but somehow this is not reflected in the current score. Based on their comments though, I am assuming the scores to be 6,8,6 which makes the cut for ICLR. Therefore, I recommend to accept this paper.